# Ceftazidime-Avibactam Treatment for *Klebsiella pneumoniae* Bacteremia in Preterm Infants in NICU: A Clinical Experience

**DOI:** 10.3390/antibiotics12071169

**Published:** 2023-07-10

**Authors:** Andrea Marino, Sarah Pulvirenti, Edoardo Campanella, Stefano Stracquadanio, Manuela Ceccarelli, Cristina Micali, Lucia Gabriella Tina, Giovanna Di Dio, Stefania Stefani, Bruno Cacopardo, Giuseppe Nunnari

**Affiliations:** 1Department of Clinical and Experimental Medicine, University of Catania, 95123 Catania, Italy; cacopard@unict.it (B.C.); giuseppe.nunnari1@unict.it (G.N.); 2Department of Biomedical and Biotechnological Sciences, University of Catania, 95123 Catania, Italy; s.stracquadanio@unict.it (S.S.); stefania.stefani@unict.it (S.S.); 3Unit of Infectious Diseases, Department of Clinical and Experimental Medicine, University of Messina, 98124 Messina, Italy; sarah.pulvirenti@libero.it (S.P.); edo.campanella93@gmail.com (E.C.); crysmica@gmail.com (C.M.); 4Unit of Infectious Diseases, School of Medicine and Surgery, Kore University of Enna, 94100 Enna, Italy; manuela.ceccarelli@unikore.it; 5Neonatal Intensive Care Unit, ARNAS Garibaldi Hospital, 95124 Catania, Italy; ltina@arnasgaribaldi.it (L.G.T.); giovannadidio@arnasgaribaldi.it (G.D.D.)

**Keywords:** ceftazidime/avibactam, NICU infections, *Klebsiella* bacteremia, infections in preterm neonates-

## Abstract

Ceftazidime/avibactam (CAZ/AVI) is an antibiotic combination approved for the treatment of several infections caused by multi-drug resistant (MDR) Gram-negative bacteria. Neonates admitted to the Neonatal Intensive Care Unit (NICU) are at high risk of developing bacterial infections, and the choice of appropriate antibiotics is crucial. However, the use of antibiotics in neonates carries risks such as antibiotic resistance and disruption of gut microbiota. This study aimed to assess the safety and efficacy of CAZ/AVI in preterm infants admitted to the NICU. Retrospective data from preterm infants with *Klebsiella pneumoniae* bacteremia who received CAZ/AVI were analyzed. Clinical and microbiological responses, adverse events, and outcomes were evaluated. Eight patients were included in the study, all of whom showed clinical improvement and achieved microbiological cure with CAZ/AVI treatment. No adverse drug reactions were reported. Previous antibiotic therapies failed to improve the neonates’ condition, and CAZ/AVI was initiated based on clinical deterioration and epidemiological considerations. The median duration of CAZ/AVI treatment was 14 days, and combination therapy with fosfomycin or amikacin was administered. Previous case reports have also shown positive outcomes with CAZ/AVI in neonates. However, larger trials are needed to further investigate the safety and efficacy of CAZ/AVI in this population.

## 1. Introduction

Ceftazidime/avibactam (CAZ/AVI) is an antibiotic combination that has been approved by the US Food and Drug Administration for the treatment of complicated urinary tract infections, complicated intra-abdominal infections, and hospital-acquired pneumonia, among other indications. This medication is particularly useful in the treatment of infections caused by multi-drug resistant (MDR) Gram-negative bacteria [1]. Ceftazidime is a potent third-generation cephalosporin with broad-spectrum activity. Its mechanism of action involves binding to penicillin-binding proteins (PBPs) and disrupting the synthesis of the bacterial cell wall, ultimately causing cell lysis and death. Avibactam is a novel non-β lactam-β-lactamase inhibitor, belonging to the class of azabicycloalkanes. While avibactam does not possess intrinsic antimicrobial activity, it enhances the efficacy of ceftazidime-avibactam by protecting ceftazidime from degradation by various serine β-lactamases. In particular, avibactam achieves its activity through covalent acylation of its β-lactamase targets, and although the process is slowly reversible, intact avibactam is released upon deacylation. Avibactam protects ceftazidime from several enzymes, effectively inhibiting Ambler class A (e.g., TEM-1, CTX-M-15, KPC-2, KPC-3), class C (e.g., AmpC), and specific class D β-lactamases (e.g., OXA-10, OXA-48), it is not active against class B enzymes (metallo-β-lactamases) [2].

CAZ/AVI showed to be superior to comparators in the treatment of bacteremia caused by carbapenem resistant Enterobacterales, especially KPC producing *K. pneumoniae* [3]. A recent meta-analysis highlighted the high and increasing rate of bacteremia caused by *K. pneumoniae*, this resulting the second most common cause of gram-negative bacteremia, following *E. coli* [4,5]. As expected, the mortality rate was higher for infections caused by ESBL or KPC producing *K. pneumoniae* regardless of the patient’s clinical characteristics [4,6]. Unfortunately, these studies often lack information about bacteremia in newborns and preterm neonates.

Infants admitted to the Neonatal Intensive Care Unit (NICU) are at high risk of developing bacterial infections due to their immature immune systems and frequent exposure to invasive procedures, such as mechanical ventilation and vascular catheterization [7]. In addition to preventive measures, hygiene, and environmental services practices, contact precautions, cohorting measure, antibiotic therapy is often necessary for the management of such infections, and the choice of the most appropriate molecule is critical to achieve successful outcomes. However, the use of antibiotics in neonates is not without risks, as it can lead to the development of antibiotic resistance and disrupt the balance of normal microbiota in the gut. In addition, some antibiotics are associated with adverse effects, including nephrotoxicity and ototoxicity [8,9]. Given the high prevalence of antibiotic-resistant infections in the NICU, there is a need for effective drugs that could be safely used in neonates. CAZ/AVI is a promising candidate due to its broad-spectrum activity against gram-negative bacteria and its favorable safety profile in adults and in pediatric patients [10]. Despite its potential benefits, there are limited information available on the safety and efficacy of ceftazidime/avibactam in neonates, particularly in those admitted to the NICU [11]. Therefore, this study aims to assess the use of CAZ/AVI in this population and evaluate its safety and efficacy in the treatment of difficult-to-treat bacterial infections.

## 2. Results

Eight patients (3 males, 5 females) with *Klebsiella pneumoniae* bacteremia received a course of CAZ/AVI and their data have been collected and analyzed.

Median gestational week was 26.5 (ranging from 24 w + 5 days to 30 w + 4 days). Median weight at birth was 940 g (ranging from 590 to 1450 g). Three infants suffered by Premature Rupture of Membranes (PROM). Median Clinical Risk Index for babies (CRIB) II score [12] at NICU admission was 9.5 (ranging from 4 to 14). All infants had a central venous catheter (CVC) and all of them received mechanical ventilation due to respiratory distress syndrome.

According to hospital protocol for neonatal sepsis prevention [13], all patients received antibiotic prophylaxis, with ampicillin plus gentamicin, for a median duration of 5.5 days (Table 1). Due to worsening of neonates’ clinical status and fever, while waiting for blood cultures results, all the patients started empirical broad spectrum antibiotic therapies for a minimum duration of 4 days.

According to blood cultures results, all the patients developed a *K. pneumoniae* bacteremia; seven isolates were ESBL-producers, one strain was carbapenem-resistant (CR) (*kpc* gene was detected) (Table 1). *K. pneumoniae* isolates resistance patterns are reported in Table 2. Only one culture from CVC tip resulted positive for *Klebsiella*.

Infants with ESBL-producers bacteremia received at least one antibiotic against *Klebsiella*, one of which was a carbapenem (imipenem or meropenem), for a total duration ranging from 4 to 34 days. Conversely, the patient with the infection sustained by CR-*Klebsiella* started CAZ/AVI immediately after blood culture results. As regards the 7 patients with ESBL-producers bacteremia, although isolated bacteria were susceptible to combination therapies according to their antibiograms, clinical conditions did not show any amelioration along with lab parameters and respiratory performance deterioration (Table 3). Based on epidemiological criteria and clinical needs, antibiotic therapy was modified switching to CAZ/AVI-based regimens.

Median duration of CAZ/AVI treatment was 14 days (ranging from 7 to 18 days). CAZ/AVI dose was 50 mg/kg (40 mg of ceftazidime plus 10 mg of avibactam) three times daily (every 8 h). CAZ/AVI was used as combination therapy along with fosfomycin for seven patients whereas amikacin was administered as companion drug in one patient. All the patients experimented clinical improvement along with lab parameters ameliorations (Table 3) until discharge. Blood cultures performed 48 h following CAZ/AVI administration resulted negative for all patients.

Concerning safety profile, none of patients developed adverse drug reactions. Median in-hospital stay was 70 days (ranging from 42 to 132 days). Median duration of mechanical ventilation was 17.5 days (from 5 to 27 days).

## 3. Discussion

The present study retrospectively analyzed the use of CAZ/AVI in preterm infants admitted to the NICU who developed *K. pneumoniae* bacteremia. The aim of the study was to evaluate the safety, clinical and microbiological efficacy of CAZ/AVI in this population. As far as we know, this is the first case series with more than two patients assessing CAZ/AVI administration in preterm infants admitted to NICU. All the patients received CAZ/AVI to treat sepsis following *Klebsiella* bacteremia not improving with broad spectrum antibiotics.

Based on epidemiological settings and due to clinical deterioration, antibiotic therapy was switched to CAZ/AVI, obtaining clinical improvement and microbiological eradication. Blood cultures results revealed that seven out of eight isolates were classified as ESBL-producing *K. pneumoniae*, while one strain exhibited carbapenem resistance. The median duration of treatment with CAZ/AVI was 14 days, and the median dose was 50 mg/kg three times daily (every 8 h), as reported by other case reports and in the pediatric dosage of the antibiotic schedule [14]. All patients received CAZ/AVI as combination therapy along with fosfomycin in seven patients and amikacin in one patient. Combination therapies with fosfomycin have been favored mostly due to the increasing frequency of CAZ/AVI-resistant strains in our settings along with the results of latest studies that highlighted the possible ecological advantage of CAZ/AVI + fosfomycin in terms of secondary infections [15,16]. For the same reasons, amikacin was added for one patient to make synergism. Colistin has not been considered as antibiotic choice due to its side effects, nephrotoxicity and neurological issues, and because there is a lack of data on its administration in newborns/infants [17].

All the patients achieved clinical and microbiological cure, as highlighted by physical conditions, parameters ameliorations and negative blood cultures. To date, only smaller case reports have been reported about CAZ/AVI administration in NICU patients. Asfour et al. [18] reported two cases of premature neonates with carbapenem-resistant Enterobacterales (CRE) infections who were treated with CAZ/AVI. In both cases, the infants had a history of respiratory distress and received multiple antibiotics prior to the diagnosis of CRE infection. In the first case, the neonate developed meningitis and bacteremia caused by carbapenem-resistant *K. pneumoniae*, which was sensitive to CAZ/AVI. This latter was added to the existing colistin therapy, resulting in clinical improvement. In the second case, the neonate had a bloodstream infection with ESBL-producing *K. pneumoniae* that became resistant to meropenem. CAZ/AVI was initiated along with amikacin and led to a significant improvement in laboratory results. Unfortunately, the second neonate died on day 61 of life. Coskun et al. [19] described a neonates who was born prematurely at 27 weeks and required intubation along with surfactant administration due to respiratory distress. He developed necrotizing enterocolitis, and then *K. pneumoniae* bacteremia treated with carbapenem plus colistin. Following that, the patient developed a pandrug-resistant *K. pneumoniae* UTI which was successfully treated with 10 days of CAZ/AVI 50 mg/kg/dose resulting in clinical amelioration and urine sterilization, confirmed at 1-month follow-up.

Iosifidis et al. [20] retrospectively analyzed nine patients treated with CAZ/AVI, four of which were infants in NICU with *K. pneumoniae* bacteremia. CAZ/AVI was administered at a dose of 62.5 mg/kg every 8 h for a total duration of treatment ranging from 4 to 38 days (median 14 days). All of patients received ceftazidime-avibactam along with at least other 3 antimicrobial agents, such as carbapenems, fosfomycin, amikacin, and colistin.

All the patients achieved clinical and microbiologic responses, and no patient died by day 30. Interestingly, one patient received two full courses of CAZ/AVI within three months, without developing resistance to this antibiotic.

Nascimento et al. [21] described a case of a 29-weeks premature infant treated with CAZ/AVI monotherapy, 50 mg/kg every 8 h for 14 days, due to a bloodstream infection caused by MDR *K. pneumoniae*, obtaining clinical and microbiological resolution.

Concerning the present study, although according to their antibiograms all the isolates should have been susceptible to carbapenem-regimens, neonates did not improve with standard therapy as confirmed by blood tests and clinical conditions. We could only speculate on possible explanations for this phenomenon. First of all, inadequate source control could have led to an incomplete bacteremia clearance with developing of antibiotic-resistant strains. In neonates it is laborious to obtain a prompt and adequate source control since it is difficult to find and maintain central-line catheters [22]. Heteroresistance, which is the presence of a subpopulation of bacteria displaying higher MIC values compared to the dominant population, could be another possible explanation for neonates’ clinical deterioration despite broad spectrum antibiotic therapies [23]. Heteroresistance in Gram-negative bacteria arises because of several mechanisms, with spontaneous tandem gene amplification being the most prevalent. Other potential causes of heteroresistance include efflux pumps overproduction and diminished expression of porins. The main driver of β-lactams resistance in *K. pneumoniae* stem from the expression of ESBLs (CTX-M, SHV), AmpC-β-lactamases (DHA, FOX), and carbapenemases (KPC). Additionally, increased resistance of *K. pneumoniae* to β-lactams can arise from deletions or mutations within the major outer membrane porins OmpK35 and OmpK36. Notably, the emergence of imipenem-resistant and meropenem-resistant subpopulations has been already reported in scientific literature [24,25,26]. Overall, the study presented here is consistent with previous reports that suggest that CAZ/AVI could be an effective and safe option for the treatment of difficult-to-treat gram-negative bacterial infections in neonates, even in empirical therapy guided by hospital epidemiology and patients’ clinical conditions.

However, it is important to note that the overall sample size in these studies is small, and larger trials are needed to further evaluate the safety and efficacy of CAZ/AVI in this delicate population. Additionally, the use of retrospective data collection may introduce bias and limit the generalizability of the study findings. Nonetheless, given the lack of clinical studies and literature data, the present study provides valuable insights into the use of CAZ/AVI in neonates and highlights the need for further research in this area.

## 4. Materials and Methods

We retrospectively collected and analyzed data about CAZ/AVI administration in preterm infants admitted to Neonatal Intensive Care Unit (NICU) in an Italian tertiary level care hospital between August 2019 and September 2022.

Medical records from neonates who were treated with CAZ/AVI were reviewed. Pharmacy data along with infectious disease consultations were used to identify all eligible patients.

CAZ/AVI was prescribed only by infectious disease consultants for the treatment of difficult-to-treat infections in patients with sepsis or septic shock.

Although identified bacterial strains were not carbapenem resistant, CAZ/AVI was empirically prescribed based upon clinical status and lack of improvement with previous antibiotic therapies, especially dealing with settings characterized of high level of carbapenem resistance as the NICU.

Information have been recorded about demographic characteristics (age, sex, weight), comorbid diseases, severity of illness upon admission, description of index infection requiring administration of CAZ/AVI (clinical, laboratory and microbiologic data), data related to administration of CAZ/AVI (dose, duration) or other antimicrobial agents (concomitant to CAZ/AVI), microbiologic and clinical responses, adverse events, and outcome. Severity was assessed using CRIB II score. Clinical response was defined as improvement/resolution of major symptoms and signs. Microbiological response was defined as evidence of negative blood culture assessed at least 48 h following CAZ/AVI administration. CVC tips from all the neonates was cultured in BHI broth (Oxoid, Basingstoke, Hampshire, UK) at 37 °C for 18 h, then the culture was streaked on blood agar to assess bacterial growth.

Blood culture positivity, identification and susceptibility tests have been performed using Beckman Coulter. Antibiotic susceptibility ore resistance was assigned in accordance with EUCAST breakpoints for Enterobacterales [27]. As EUCAST guidelines, fosfomycin susceptibility tests were performed by agar dilution using the AD Fosfomycin 0.25–256 ready to use panel (Liofilchem, Teramo, Italy) according to manufacturer’s protocol.

Detection of carbapenem-resistance gene (kpc) was performed using Xpert^®^ Carba-R—Cepheid according to manufacturer instructions [28].

## Figures and Tables

**Table 1 antibiotics-12-01169-t001:** Clinical characteristics of the eight premature neonates. GW: Gestational week; PROM: Premature Rupture of Membranes; CRIB: Clinical Risk Index for babies; A/G: Ampicillin/Gentamicin; BC: Blood culture; LOS: Length of stay; Disch: Discharged.

Patient ID	SEX	GW	Weight at Birth (g)	PROM	CRIB II SCORE	A/G Prophylaxis (Duration in Days)	Mechanical Ventilation (Duration in Days)	Days from Hospital Admission to First Positive BC	ESBL	KPC	LOS (Days)	Duration of CAZ/AVI (Days)	BC after 48 h of CAZ/AVI	Clinical Outcome
1	M	25 w 0 g	590	YES	14	YES (5)	YES (20)	35	X		132	14	NEG	Disch
2	M	25 w 3 g	940	NO	11	YES (7)	YES (15)	19		X	63	15	NEG	Disch
3	F	25 w 3 g	590	YES	12	YES (7)	YES (27)	7	X		106	7	NEG	Disch
4	F	28 w 4 g	1230	NO	6	YES (9)	YES (11)	16	X		67	14	NEG	Disch
5	F	29 w 4 g	1260	NO	8	YES (6)	YES (13)	22	X		73	15	NEG	Disch
6	F	24 w 5 g	650	NO	12	YES (5)	YES (26)	8	X		119	11	NEG	Disch
7	M	29 w 5 g	1450	NO	4	YES (3)	YES (22)	4	X		42	18	NEG	Disch
8	F	30 w 4 g	940	SI	8	YES (4)	YES (5)	13	X		49	11	NEG	Disch

**Table 2 antibiotics-12-01169-t002:** Antibiotic susceptibility profiles of the *K. pneumoniae* strains isolate from neonatal blood cultures.

Patient ID	1		2		3		4		5		6		7		8	
**Microorganism**	*K. pneumoniae* ESBL		*K. pneumoniae* KPC		*K. pneumoniae* ESBL		*K. pneumoniae* ESBL		*K. pneumoniae* ESBL		*K. pneumoniae* ESBL		*K. pneumoniae* ESBL		*K. pneumoniae* ESBL	
**Isolation period**	25 September 2020		31 July 2020		30 July 2020		15 September 2019		23 August 2021		1 August 2022		4 October 2022		11 October 2022	
**Sample**	Blood															
**Antibiotics**	MIC	S/R	MIC	S/R	MIC	S/R	MIC	S/R	MIC	S/R	MIC	S/R	MIC	S/R	MIC	S/R
Ampicillin	>8	R	>8	R	>8	R	>8	R	>8	R	>8	R	>8	R	>8	R
Cefotaxime	>32	R	>32	R	>32	R	>32	R	>32	R	>32	R	>32	R	>32	R
Ceftazidime	16	R	>32	R	32	R	16	R	2	S	2	S	4	S	4	S
Ciprofloxacin	1	R	1	R	>1	R	>1	R	>1	R	>1	R	>1	R	>1	R
Colistin	≤2	NC	≤2	NC	≤2	NC	≤2	NC	≤2	NC	≤2	NC	≤2	NC	≤2	NC
Ertapenem	≤0.12	S	>1	R	≤0.12	S	≤0.12	S	≤0.12	S	≤0.12	S	≤0.12	S	≤0.12	S
Gentamicin	>4	R	>4	R	>4	R	>4	R	>4	R	>4	R	>4	R	>4	R
Levofloxacin	0.5	R	>1	R	1	R	1	R	0.5	R	0.5	R	0.5	R	0.5	R
Piperacillin	>16	R	>16	R	>16	R	>16	R	>16	R	>16	R	>16	R	>16	R
Tigecycline	≤1	S	1	S	1	S	≤1	S	≤1	S	≤1	S	≤1	S	≤1	S
Amikacin	≤8	S	>16	R	≤8	S	≤8	S	≤8	S	16	S	≤8	S	≤8	S
Amoxi/clav	>32/16	R	>32/16	R	>32/16	R	16/8	R	>32/16	R	>32/16	R	>32/16	R	>32/16	R
Aztreonam	>4	R	>4	R	>4	R	4	S	>4	R	4	S	>4	R	>4	R
Cefepime	>8	R	>4	R	>4	R	4	S	>8	R	4	S	>8	R	4	S
Fosfomycin	2	S	1	S	1	S	2	S	0.5	S	2	S	4	S	4	S
Imipenem	≤1	S	>8	R	1	S	≤1	S	≤1	S	≤1	S	≤1	S	≤1	S
Meropenem	≤0.12	S	>8	R	≤0.12	S	≤0.12	S	≤0.12	S	≤0.12	S	≤0.12	S	≤0.12	S
Pip/tazo	≤8	S	>16	R	≤8	S	16	S	≤8	S	16	S	>16	R	≤8	S
Tobramycin	>4	R	>4	R	>4	R	>4	R	>4	R	>4	R	>4	R	>4	R
TMP/STX	4/76	S	2/38	S	2/38	S	<2/38	S	>4/76	R	>4/76	R	>4/76	R	>4/76	R
CAZ/AVI	≤2	S	≤2	S	≤2	S	≤2	S	≤2	S	≤2	S	≤2	S	≤2	S
Cefto/tazo	≤1	S	≤1	S	≤1	S	≤1	S	≤1	S	≤1	S	≤1	S	≤1	S

**Table 3 antibiotics-12-01169-t003:** Lab examinations at different timepoints. T1: Laboratory values at the time of CAZ/AVI administration; T2: Laboratory values at 48 h from the start of CAZ/AVI; T3: Laboratory values at the time of CAZ/AVI cessation. LP: Laboratory parameters; WBC: white blood cells; Ne: Neutrophils; Ly: Lymphocytes; PLT: Platelets; CRP: C-Reactive Protein; PCT: Procalcitonin.

Patient ID	1			2			3			4			5			6			7			8		
LP (Reference Range)	T1	T2	T3	T1	T2	T3	T1	T2	T3	T1	T2	T3	T1	T2	T3	T1	T2	T3	T1	T2	T3	T1	T2	T3
WBC, cells/mmc × 10^3^ (9–34)	24	18	5.2	3.3	4.1	11.9	22.9	13	21	15.3	21.2	20.7	10.1	26.7	6.2	46.1	15.2	26.4	53	25.2	7	6.1	32	9.2
Ne, % (40–75)	81	54.4	36.9	55.7	35.3	49.2	36.9	23.1	64.3	81	64.1	64.3	72.1	80.2	44	60	47.1	66.3	87	65.3	39.9	38.6	62.5	31.1
Ly, % (25–50)	10	23.6	29.9	14.8	26.6	39.5	29	39.7	23.9	1	16.9	23.9	24.4	14.8	38.9	19.8	22.5	23.4	12	22.1	42.7	52.7	21.6	48.5
PLT, cells/mmc × 10^3^ (150–400)	15	53	52	9	9	44	158	276	348	7	10	100	11	9	0,1	326	37	318	14	9	71	10	41	46
CRP, mg/dL (0–0.5)	31.9	10.9	1.2	14.6	12.3	0.1	0.3	0.4	0.1	14.1	10.6	3.3	17.7	8.6	0.5	1.4	4.4	0.1	11.6	12.2	0.6	10.6	12.5	0.1
PCT, ng/mL (<0.5)	10.4	10.9	0.4	32.2	12.3	0.5	2.12	1.05	0.2	47.1	21.6	0.5	49.9	23.2	0.4	0.8	20.1	0.3	74.5	73.2	1.1	5.1	3.05	0.1

## Data Availability

The data presented in this study are available on request from the corresponding author.

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
