# Peer review of "Ceftazidime-Avibactam Treatment for Klebsiella pneumoniae Bacteremia in Preterm Infants in NICU: A Clinical Experience"

_antibiotics, 2023, doi:10.3390/antibiotics12071169_

Round 1
Reviewer 1 Report
Dear Sir,
I found this article important to the pediatric field, especially for the newborn population, highlighting the importance of this report. I am sending some considerations that could improve your article.
Results:
Include information on the primary or secondary source of the bacteremia and if any other microorganism was isolated previously to the K.pneumoniae report.
You should include the abbreviatures used in Table 1.
Discussion:
You said that based on epidemiological evidence suggests the use of CAZ/AVI, but there is no more information about the microorganism. Colistin is approved for bacteremia but you did not include this antibiotic in your treatment, why?
Dear Sir,
I found this article important to the pediatric field, especially for the newborn population, highlighting the importance of this report. I am sending some considerations that could improve your research.
Results:
Include information on the primary or secondary source of the bacteremia and if any other microorganism was isolated previously to the K.pneumoniae report, including other infection or colonization.
You should include the abbreviatures used in Table 1.
Discussion:
You said that based on epidemiological evidence suggests the use of CAZ/AVI, but there is no more information about the microorganism. Colistin is approved for bacteremia but you did not include this antibiotic in your treatment, why?
Author Response
Thank you for your kind words. As regards the points you raised, Klebsiella was the first and the only pathogen isolated from infants. Concerning the source of bacteremia, it wasn’t easy to clarify this point, since it was not practical to perform adequate and prompt source control in neonates. However, we added some extra information in the results and in the discussion section. About colistin administration, we stated in the text that it has not been considered as antibiotic choice due to its side effects, nephrotoxicity and neurological issues, and because there is a lack of data on its administration in newborns/infants. Finally, we added abbreviations in table 1 as you suggested.
Reviewer 2 Report
This retrospective study examines the use of ceftazidime-avibactam (CAZ/AVI) among preterm infants who were hospitalized in neonatal intensive care units (NICUs) due to Klebsiella pneumoniae bacteremia. The objective was threefold: firstly. We aimed at determining whether administering CAZ/AVI would result in improved patient safety; secondly. We assessed if there were indications of clinical effectiveness and finally. We evaluated microbiological outcomes. This study represented the first case series involving more than two patients assessing CAZ/AVI administration in preterm infants admitted to NICU. All septic patients with Klebsiella bacteremia, who failed to improve using broad spectrum antibiotics. Were treated using CAZ/AVI.
Line 17, 39 - Gram-negative is capitalized.
Line 22- "Klebsiella (K.) pneumoniae" - Remove the "k.".
During the entire introduction, they didn't mention Klebsiella once. This makes the intro too poor.
An introduction with only 7 references is too little.
The tables are too confusing and unappealing. The authors have to drastically change the presentation.
Line 92 - unformatted.
The sample of this work is very small.
This work is too clinical for the journal in question. I think that more studies will be needed, as well as a larger sample.
Author Response
Line 17, 39 - Gram-negative is capitalized.
Reply: We corrected the typo.
Line 22- "Klebsiella (K.) pneumoniae" - Remove the "k.".
Reply: We changed the text as you suggested
During the entire introduction, they didn't mention Klebsiella once. This makes the intro too poor.
Reply: Thank you for your precious suggestions. We added few lines about what you pointed out along with appropriate references.
An introduction with only 7 references is too little.
Reply: We added extra appropriate references in the introduction section.
The tables are too confusing and unappealing. The authors have to drastically change the presentation.
Reply: Thank you for your opinion. We reformatted the tables, especially the one with antibiotic susceptibilities. I think there was a mistake with the file upload procedure since the tables were differently formatted in our previous version.
Line 92 - unformatted.
Reply: We changed what you suggested.
The sample of this work is very small.
This work is too clinical for the journal in question. I think that more studies will be needed, as well as a larger sample.
Reply: As far as we know this is the biggest case series of preterm infants treated with CAZ/AVI.
We strongly agree with you about the small sample size, and we are working to collect other multicentric cases to make future bigger studies with statistical analysis and stronger results. Also, it is not so easy to work with preterm neonates as regards data collecting and clinical management.
We took the journal Antibiotics in consideration as we would like to raise the attention on the possible administration of CAZ/AVI in this special population.
Reviewer 3 Report
The paper by Marino et al aims to asssess the safety and efficacy of CAZ/AVI in preterm infants admitted to the NICU. Taking into account wide spread of resistance to antimicrobials, the development of novel approaches to inhibit pathogenic bacteria (may be also foodborne ones) is an important challenge over the world and the topic is very important for the clinical practice. The work scientifically sounds and can be considered for publication. The reviewer thinks that the paper belongs rather to short communication than the full article.
Author Response
Thank you for your kind words. We rephrase the title talking about “case series” rather than “retrospective study”.
Round 2
Reviewer 1 Report
Dear Sir,
The article is relevant for the field.
Best regards,
Reviewer 2 Report
The authors took into account all proposed changes and the article was substantially improved.